# Targeting Myeloid-Derived Suppressor Cell Trafficking as a Novel Immunotherapeutic Approach in Microsatellite Stable Colorectal Cancer

**DOI:** 10.3390/cancers15225484

**Published:** 2023-11-20

**Authors:** Benny Johnson

**Affiliations:** Department of Gastrointestinal Medical Oncology, Division of Cancer Medicine, The University of Texas MD Anderson Cancer Center, Houston, TX 77030, USA; bjohnson6@mdanderson.org or benjohnson2010@gmail.com

**Keywords:** microsatellite stable colorectal cancer, myeloid-derived suppressor cells, immunotherapy, tumor microenvironment, minimal residual disease

## Abstract

**Simple Summary:**

Microsatellite stable colorectal cancer represents the majority of patients who have been diagnosed with colorectal cancer, and it is inherently resistant to currently available immunotherapy. After standard-of-care chemotherapy, patients have very limited remaining treatment options. Therefore, defining innovative approaches to reactivate the immune system is critical, and novel treatments are desperately needed. Here, we review and highlight a key immune cell that is involved in reprogramming the immune system in colorectal cancer called the myeloid-derived suppressor cell. The successful targeting of myeloid-derived suppressor cells with novel drug combinations in clinical trials may allow for microsatellite stable colorectal cancer to become responsive to immunotherapy and thereby improve patients’ outcomes.

**Abstract:**

Myeloid-derived suppressor cells (MDSCs) are a unique subset of immune cells that promote an immunosuppressive phenotype due to their impacts on CD8 and regulatory T cell function. The inhibition of MDSC trafficking to the tumor microenvironment (TME) may represent a novel target in microsatellite stable (MSS) colorectal cancer with the potential to reprogram the immune system. Here, we review the rationale of inhibiting myeloid suppressor cell trafficking in treatment-refractory MSS colorectal cancer and circulating tumor DNA (ctDNA) positive settings to determine whether this approach can serve as a backbone for promoting immunotherapy response in this difficult-to-treat disease.

## 1. Introduction

Colorectal cancer (CRC) is the second leading cause of cancer-related deaths in the United States, with over 150,000 patients diagnosed in 2022, 25% of whom present with metastatic disease [1]. The five-year survival rate continues to be dismal at less than 15% [1]. Unfortunately, only 4% of patients with mCRC harbor microsatellite instability high (MSI-H) colorectal cancer, rendering immune checkpoint blockade (ICB) ineffective for the majority of patients with microsatellite stable (MSS) disease. Furthermore, CRC, which has traditionally impacted older age patients (the median age in the early 2000s was 72 years old), has taken a dramatic epidemiologic shift towards a younger patient population (median age of diagnosis = 66 years old) over the past several decades [2]. By 2030, it is estimated that the incidence rates for colon and rectal cancer will increase by 90% and 124.2%, respectively, for patients aged 20–34 years old [3]. Additionally, our group has shown that early-onset CRC (EOCRC) patients with *RAS*-mutated tumors treated with neoadjuvant chemotherapy and liver resection tend to have worse overall survival, highlighting that “traditional” chemotherapy and surgical approaches for this group remain quite limited [4]. Therefore, innovative approaches for patients with treatment-refractory *RAS*-mutated MSS CRC are desperately needed.

## 2. MDSC Immunosuppression: A Critical Mechanism of Immune Evasion in Cancer

Myeloid cells represent the most abundant nucleated hematopoietic cells in the human body and are a collection of distinct cells with both immunosuppressive and immunostimulatory functions [5]. Of note, myeloid cells are heterogenous, and they do not all arise from circulation [6]. Innately, myeloid cells can protect organisms from pathogens, eliminate dying cells, and mediate tissue remodeling. The three groups of terminally differentiated myeloid cells (i.e., macrophages, dendritic cells (DCs), and granulocytes) play critical roles in the normal function of both the innate and adaptive immune systems [5]. Myeloid cells have the ability to differentiate into varying phenotypes based upon the signals received from the tumor microenvironment (TME) [5], representing an immensely complex process in how they relate to cancer biology, most notably promoting an underlying resistance to cancer treatments. Cancers pathologically “hijack” and alter myeloid cells into potent immunosuppressive cells that include myeloid-derived suppressor cells (MDSCs) and neutrophils, facilitating tumor cells to successfully evade the immune system [5]. Pathologic myeloid cell compartment activation occurs due to persistent stimulation, secondary to inflammatory signals and myeloid growth factors such as GM-CSF, macrophage colony-stimulating factor (M-CSF), IL-6, and IL-1B [5].

Therefore, one hypothesis of the reason for the overall lack of ICB efficacy in patients with MSS mCRC is believed to be an immunosuppressive barrier orchestrated by myeloid cells in the TME, which include myeloid-derived suppressor cells (MDSCs) and neutrophils [7]. Myeloid cells are generally immunosuppressive via the promotion of Foxp3+ Tregs, the promotion of reactive oxygen and nitrogen species, metabolic competition, and immunosuppressive cytokines [7] (Figure 1). Furthermore, it is suspected that the underlying resistance to chemotherapy in GI cancers, specifically colorectal cancer, may be due to the escape of immune surveillance acquired by the activation of cell populations that promote immunosuppression, such as MDSCs, as well as regulatory T cells and tumor-associated macrophages promoting metastasis [8,9,10].

MDSCs are present in both the primary tumor and metastatic sites inhibiting T cell-mediated and natural killer (NK) cell-mediated tumor clearances [11,12]. Chemokines secreted by tumors recruit myeloid cells into the TME by binding to the chemokine receptors CXCR1 and CXCR2 (“CXCR1/2”) on the myeloid cell surface [13] (Figure 1). Preclinical models have demonstrated that *RAS* mutations in mCRC (*KRAS* or *NRAS*) drive chemokine secretion by tumor cells via the KRAS-IRF2 axis, thereby promoting an “immune cold” TME with a resultant de novo resistance to immunotherapy [13] (Figure 1). Furthermore, the tumor-mediated impact on the differentiation of myeloid cells is not only confined to the tumor; rather, it is a systemic phenomenon. Chemokines released by the tumor act at distant organ sites that include the bone marrow and spleen, where they alter DCs, macrophages, and granulocytes to become immunosuppressive and accumulate within the tumor [7].

The process of MDSC development is believed to occur in two phases, with the first being myeloid cell expansion occurring in the bone marrow and the spleen. The second phase is the conversion of neutrophils and monocytes to pathologically active MDSCs, primarily occurring in the peripheral tissues [7]. MDSCs are critical players in the TME of malignancy due to their abilities to function as inhibitors of immune responses from T cells, B cells, and natural killer (NK) cells [7] (Figure 1). Therefore, MDSCs may promote cancer metastases via four mechanisms: (i) the evasion of tumor cells from immune-mediated clearance, (ii) the remodeling of the TME (e.g., angiogenesis and tumor invasion), (iii) being key cells that are involved in the premetastatic niche, and (iv) communication with tumor cells to promote the epithelial-to-mesenchymal transition [14]. MDSCs appear to play key roles in the suppression of the immune system, and this occurs via the targeting and release of various soluble factors that impact the immune response [14]. Most notably, the expressions of arginase (ARG1), COX2, ROS, transforming growth factor β (TGFβ), IL-10, inducible NOS (iNOS), and NADPH oxidase 2 (NOX-2); the sequestration of cysteine, programmed death-ligand 1(PD-L1), indoleamine 2,3-dioxygenase (IDO); and the induction of T regulatory cells all directly impact cytotoxic T cells and other relevant immune cells such as NK cells [15]. Furthermore, elevated ROS levels in the microenvironment of colon cancer has been shown to downregulate the response of antigen-specific T cells. [15].

Over the years, we have gained an increasing appreciation of the complexity of the myeloid compartment and its impact on tumor growth by facilitating inherent resistance to cancer treatments, specifically immunotherapy. The trafficking of myeloid cells into the TME has known prognostic and therapeutic implications in cancer [5], with a recent focus on the biology of chemokine and cytokine pathways such as CXCR2, CCR5, and CSF1R [5]. Furthermore, the proximity of myeloid cells to tumor cells has been shown to have prognostic significance in pancreatic cancer [16]. Therefore, considering the immense immune cell infiltrates in GI cancers, specifically CRC, impeding trafficking to the TME may improve therapeutic efficacy and represent the backbone for subsequent combinatorial strategies. While there are many potential avenues to target MDSCs’ roles in cancer immunology (i.e., MDSC depletion and MDSC reprograming), here, we focus on myeloid recruitment into the tumor microenvironment of CRC and whether impeding this function therapeutically can serve as a foundational approach to enhance the immune response in MSS colorectal cancer.

## 3. CXCR1/2 Are Critical in the Tumor-Driven Recruitment of MDSCs to the Tumor

The G protein-coupled receptors, CXCR1 and CXCR2, located on MDSCs are crucial in the tumor-driven recruitment of MDSCs to the tumor and thereby represent an attractive target for novel immunotherapeutic approaches. Tumors secrete the soluble chemokines, CXCL1, CXCL2, CXCL3, CXCL5, CXCL6, CXCL7, and/or CXCL8, that cause MDSCs in the bone marrow and elsewhere to become trafficked to the tumor [17]. These chemokines all activate CXCR2. The CXCL8 (IL-8) and CXCL6 chemokines also recruit pro-tumoral myeloid cells to tumors by activating the closely related isoform, CXCR1 [18]. Thus, dual CXCR1/2 inhibition, as provided by SX-682, is necessary to completely inhibit this axis and may represent an ideal strategy. Pre-clinical data show that disrupting the CXCR2-MDSC-tumor axis (via CXCR inhibition or CXCR gene deletion in MDSCs) in mouse tumor models is therapeutically efficacious across a wide variety of solid tumor types that are exclusive to the variable genetic alterations within the tumor itself [19,20,21,22,23,24,25,26].

CXCR2 inhibition in mice results in the abrogation of the tumor’s “cloaking device”, resulting in (i) the blockade of MDSC recruitment to the tumor, (ii) the blockade of metastasis, (iii) the blockade of local invasion, and (iv) the blockade of immune escape and growth [23,24]. Therapeutically, the inhibition of CXCR2 in these models is synergized with both conventional chemotherapy and anti-PD1 immunotherapy. Anti-tumor immunotherapy via the CXCR1/2 blockade is therefore not limited in principle to any single tumor type, but it may have activity in augmenting a therapeutic immune response to a number of histologically distinct tumors. In this review, we highlight its potential utility in colorectal cancer. Refractory *RAS*-mutated (*KRAS* or *NRAS*) MSS mCRC represents a significant unmet need with no response to ICB monotherapy, and novel approaches to reprogram the immune tumor microenvironment are imperative. Preclinical models have determined that *KRAS* mutations themselves drive the CXCR2 axis and support an immune-suppressive phenotype by upregulating the CXCL3 chemokine, and they promote CXCR2+ MDSCs to the TME [13].

## 4. Core *RAS* Mutations in mCRC (*KRAS* or *NRAS*): Drivers of CXCR2 Chemokine Secretion by Tumor Cells

KRAS is a phosphorylated signal transducer that, under normal physiologic conditions, self-inactivates via intrinsic guanosine triphosphatase (GTPase) activity. In CRC, several *KRAS* oncogene mutations have been identified, initially in exon 2 (codons 12 and 13), resulting in the production of proteins with reduced GTPase activities. The frequency of *KRAS* exon 2 mutations in CRC is reported to be about 40%. This is clinically significant for patients, as the presence of *KRAS* mutations preclude the use and efficacy of anti-epidermal growth factor receptor (EGFR) antibodies, such as cetuximab and panitumumab. In 2013, it was established that low-frequency mutations in *KRAS* exon 3 and 4 and *NRAS* (all *RAS* testing) also promote underlying resistance to anti-EGFR therapy, reflecting an additional 10–15% of patients beyond the initially reported 40% of patients with traditional *KRAS* exon 2 mutations. Therefore, greater than 50% of patients with mCRC have an expanded *RAS* mutation, correlating with disease aggressiveness and obviating the option of biomarker-driven anti-EGFR antibody therapy. Recently, there has been an exciting development of *RAS* inhibitors targeting *KRAS* G12C mutations in lung and colorectal cancer with the combination of a *KRAS* G12C inhibitor and anti-EGFR antibody, resulting in meaningful response rates for patients with refractory mCRC [27,28]. While ongoing efforts are underway to target additional *KRAS* mutations more globally, currently, only 2–3% of mCRC patients have a *KRAS* G12C mutation, relegating the use of this targeted approach to a very rare subset [29]. Considering this, an unmet need remains for innovative treatment approaches in *RAS*-mutated MSS mCRC, especially those that involve novel immunotherapeutic strategies.

## 5. Mechanism of the *KRAS*-Promoted Immunosuppressive Tumor Microenvironment

iKAP (induced KRAS, APC, and TP53) represents a novel CRC mouse model that appropriately reflects the progression in human tumors [13]. A pathway analysis of the RNA-seq profiles from iKAP tumors revealed that *KRAS*-mutated tumors showed the downregulation of interferon γ (IFN γ) and interferon alpha (IFN α) responses as the most significant pathways. Since the loss of IFN responses reflects the suppression of tumor immune surveillance, computational analyses were pursued to identify a gene or set of genes that may represent the *KRAS*-mediated suppression of IFN responses [13]. This identified a single gene, interferon regulatory factor 2 (IRF2), a transcription factor that binds to the IFN-stimulated response element, with the IFN consensus sequence and IRF element playing critical roles [13]. iKAP tumors demonstrated that low or negative IRF2 expression is found in KRAS-mutated tumors. In the iKAP model, KRAS extinction resulted in decreased p-ERK expression and increased IRF2 expression, suggesting that the repression of IRF2 may reflect a novel mechanism of underlying KRAS-driven immune suppression in the tumor microenvironment of mCRC [13]. CXCL3 is a member of the IL-8 angiogenic cytokine family, secreted by monocytes, macrophages, and cancer cells, and it is directly involved with the chemotaxis and cell activation of neutrophils [13]. CXCR2 is the cognate receptor for CXCL3 and is critical for MDSC migration from bone marrow to tumors via an interaction with tumor-secreted ligands [13]. CXCL3 harbors IRF2 binding elements in its promoter with enhanced binding upon IRF2 overexpression. Real-time qPCR and ELISA assays revealed that CXCL3 expression and secretion were significantly suppressed with enforced IRF2 expression or following KRAS extinction in iKAP cells [13]. This suggests that there may be a key role of the CXCL3/CXCR2 axis in modulating the immune microenvironment in MSS mCRC. These data support CXCL3-CXCR2 as a critical effector of the KRAS-mutated IRF2 axis and provides an actionable approach to inhibit MDSC infiltration and enable T cell-mediated tumor immunity in CRC [13]. SX-682 is a novel oral small-molecule immunotherapy that disrupts CXCR1/2 signaling [30,31,32,33]. By removing this immunosuppressive cloak, SX-682 is predicted to unmask the greater efficacy of immunotherapies (e.g., anti-CTLA4 and anti-PD1), whose mechanisms are complementary and non-redundant to the mechanism of SX-682. We therefore aimed to test the hypothesis that the combination of SX-682 with anti-PD-1 will demonstrate anti-tumor activity with adequate safety and tolerability for patients with refractory *RAS*-mutated (*KRAS* or *NRAS*) MSS mCRC in our ongoing investigator-initiated clinical trial (NCT04599140).

The combination of a dual immune checkpoint blockade (ICB) with anti-CTLA4 and anti-PD-1 has provided durable responses for a subset of patients with chemotherapy-refractory tumor types, leaving those left in the balance desiring similar outcomes to be achieved [5]. Unfortunately, many malignancies demonstrate an innate resistance to ICB, while others develop adaptive or secondary resistance, preventing meaningful responses [5,34]. It is believed that in cancers such as MSS colorectal cancer, the presence of myeloid cells represents one critical mechanism for the development of a resistance to ICB [5,35,36]. As previously summarized, the infiltration of polymorphonuclear and monocytic MDSCs suppress innate T cell function via both indirect and direct effects that alter the immune tumor microenvironment and are associated with poor clinical outcomes [5,37,38].

Additional KRAS mutant preclinical models in both pancreas and lung adenocarcinoma also support significant roles for MDSCs in the tumor microenvironment [39,40]. The inducible oncogenic KRAS mouse model for pancreatic ductal adenocarcinoma (PDAC) is designated as “iKRAS” and is found to have an abundance of MDSCs [39]. The clinical data for PDAC patients revealed a high frequency of MDSCs when compared to low levels, are associated with advanced disease and poor survival outcomes. Due to the known CXCR2 expression and its role in MDSC recruitment, SX-682 was used to investigate the impact of the MDSC recruitment blockade. Treatment with SX-682 in the iKRAS model revealed a notable decrease in the trafficking of MDSCs and inhibited tumor growth and improved survival in iKRAS mice [39]. In a preclinical KRAS G12D lung mouse model, it was shown that neutrophils played key roles in promoting an immunosuppressive tumor microenvironment and disease progression. Depletion studies confirmed that neutrophils promote tumor growth and reduce T cell infiltration, impacting the anti-PD-1 response as well as altering angiogenesis [40].

Due to these effects, many clinical trials have focused on targeting MDSCs to overcome this inherent resistance to immunotherapy [41,42]. The approach used to accomplish this goal can vary based on the specific strategy pursued— such as (1) altering the myeloid cell function via activation or reprogramming; (2) depleting myeloid cells primarily through impeding recruitment and thus decreasing the infiltration to the TME; or (3) promoting phagocytosis [5,42]. Here, we highlight a novel approach to target myeloid cell trafficking and whether the inhibition of this process can reverse the underlying resistance to checkpoint blockade in MSS colorectal cancer.

Interestingly, previous studies have shown that compensatory pathways emerge when the blockade of the CSF1R recruitment pathway was attempted; specifically, the CXCR2-mediated axis resulted in increased MDSCs and T regulatory cell infiltration in the TME [5,43,44]. Therefore, we suspect that the alternative may represent an escape mechanism here, where the CSF1R axis may reflect a possible compensatory pathway by increasing the recruitment of proinflammatory macrophages to the TME, resulting in an adaptive resistance to our approach of targeting the CXCR2 recruitment pathway. Such redundant pathways could impact the efficacy in MDSC recruitment strategies and will need to be considered in subsequent treatment approaches supporting a multifaceted immunotherapeutic approach for MSS colorectal cancer [5]. Of note, increased levels of CXCL8 (IL-8), a ligand for CXCR1 and CXCR2, has been associated with a decreased efficacy to immunotherapy due to the increased recruitment of MDSCs, and macrophages in various malignancies highlight that the IL-8 blockade may represent a valid combinatorial target for checkpoint inhibition in addition to the CXCR1/CXCR2 receptor itself [45,46].

It still remains to be deciphered if MDSCs play predominant roles in maintaining the immunosuppressive TME in CRC and what roles monocytes and macrophages may play in a combinatorial fashion. Are all myeloid cells involved or are some subtypes more involved than others? Myeloid cells are likely too broad a category to truly describe the complex interplay between immune cells and tumor cells at the level of the microenvironment in MSS colorectal cancer. Furthermore, it should be noted that targeting the trafficking of myeloid cells only represents one approach, as summarized here. As mentioned previously, altering the myeloid cell function via activation or reprogramming is also a viable approach. One notable example is the preclinical work, revealing that neutrophils can be activated to target tumors and reduce metastatic seeding due to the tumor necrosis factor, the CD40 agonist, and a tumor-binding antibody [47]. Complement C5a activates neutrophils to generate leukotriene B_4_-producing reactive oxygen species and oxidative damage, thereby promoting the T cell-mediated clearance of tumors via an inflammatory pathway [47]. Ultimately, a combinatorial strategy of ICT in conjunction with myeloid targeting strategies that include the depletion, reprogramming, and enhancement of phagocytosis may provide the most viable approach to combat cancer progression or mortality.

In a preclinical CRC model, the use of a CXCR2 inhibitor, SX-682, produced a modest increase in survival, while the combination of SX-682 and anti-PD1 treatment significantly extended survival for iKAP mice [13]. This combination strategy significantly increased the CD8+/Treg ratio over single agent arms. These results were also seen in the MC38K tumor model. These data support the hypothesis that MDSCs contribute to the de novo resistance of *RAS*-mutated CRC to ICB therapy and that targeting the CXCL3/CXCR2 may enhance ICB efficacy through the reduction in MDSCs in the tumor microenvironment. Based on this high-impact science generated within our institution, a single-center investigator-initiated clinical trial utilizing the novel combination of SX-682, an orally bioavailable small-molecule inhibitor of myeloid-derived suppressor cells, with nivolumab for patients with refractory *RAS*-mutant MSS CRC or circulating tumor DNA positive CRC is underway (NCT04599140, STOPTRAFFIC-1). To date, this protocol has identified the recommended phase 2 dose and is currently enrolling both the expansion Arm A and Arm B ctDNA positive cohort [48,49]. Arm B is for patients with a positive circulating tumor DNA (ctDNA) status (*n* = 15). The presence of circulating tumor DNA (ctDNA) after the completion of planned curative therapies for patients with MSS CRC identifies subclinical metastatic CRC (mCRC) patients who are characterized as positive MRD patients (minimal residual disease). Patients with positive ctDNA have micrometastatic disease-shedding tumor DNA and manifest poor survival outcomes, representing a clear unmet need. Additionally, current ctDNA assays are able to successfully identify patients with minimal residual disease, nearly 100% of whom will eventually have a radiographic recurrence. Of note, the lead time of ctDNA positivity to radiographic recurrence is up to 6–9 months, facilitating a window for the early initiation of a novel therapy with a goal towards the eradication of micrometastatic disease [50].

## 6. Minimal Residual Disease (MRD) in Colorectal Cancer Represents an Innovative Space for Early Therapeutic Intervention: Can We Target MDSCs?

The previously summarized data above are especially provocative in the context of the MRD setting. Applying this targeted immunotherapeutic approach to MRD makes sense as it represents what we believe to be an immature tumor microenvironment (TME), and considering the immunosuppressive nature that MDSCs bring to the TME, augmenting TMEs in the liver and lung “niches” to increase T cell immunity appears to be a viable approach to address MRD. After patients complete curative therapy for their primary tumor, there is rationale to support that cancer cells can enter a dormant life cycle by occupying a specific niche [51]. Upon the engagement of cells with receptors in the niche, these dormant cancer cells undergo cell cycle arrest and reprogramming to adapt to the niche and survive [51]. One aspect of this includes dormant cancer cells activating immune evasion mechanisms to hide from the immune system and promote long-term dormancy [51]. Dormant cancer cells upregulate major histocompatibility complex class II and programmed cell death 1 ligand 1 (PDL1), which may engage co-inhibitory receptors on natural killer (NK) cells and T cells and contribute to adaptive immune resistance to immune checkpoint inhibitor therapy [51]. These cells are subsequently reactivated by changes in the niche, and they proliferate and manifest as metastatic relapse, which can now be theoretically identified via ctDNA assays.

The preclinical data support that MDSC may aid in preparing distant organs for seeding via metastatic cells, thereby playing a role in establishing a premetastatic niche [52]. Considering that regulatory T (Treg) cell depletion may remove the immune privilege provided by the niche and “decloak” dormant cancer cells for immune elimination, deploying a therapy that inhibits MDSC trafficking to this immature TME may further decrease T regulatory infiltration, increase CD8 T cell infiltration, and provide an avenue for immune clearance when SX-682 is combined with nivolumab.

MDSCs appear to confer some level of protection to circulating tumor cells and appear to be involved in the formation of pre-metastatic niches (PMNs), which encourage the invasion of cancer cells [53]. MDSCs promote tumor metastasis through the reorganization of the basal membrane via the secretion of matrix metalloproteinases (MMPs), protecting circulating tumor cells (CTCs) and playing a major role in CTC clusters, which are known to promote immune escape, thereby providing a pathway for tumor cell migration, extravasation, and invasion [51,52,53] (Figure 2). MDSCs facilitate these malignant processes through the release of neutrophil extracellular traps (NETs), promoting tumor cell survival via the ROS-nuclear erythroid 2-related factor 2 (NRF2)/antioxidant responsive element (ARE) axis and Notch signaling pathway [54,55,56].

Another notable preclinical study involves the all-induced *villinCre^ER^ Kras^G12D/+^ Trp53^fl/fl^ Rosa26^N1icd/+^* (KPN) genetically engineered mouse model that manifests intestinal adenocarcinoma, followed by subsequent metastases to the liver, lymph nodes, and lungs [57]. In this study, neutrophil-associated genes such as MPO and CXCR2 were noted to be highly expressed in human CRC, specifically consensus molecular subtype 4 (CMS4). CMS4 represents a stratification of CRC based on transcriptional profiling, and it is characterized by a “mesenchymal” phenotype driven by TGF-ß signaling with intense innate immune cell and fibroblast infiltration [58]. When KPN mice were treated with AZD5069, a CXCR2 small-molecule inhibitor, neutrophil recruitment was blocked to the peripheral blood and primary tumors with an increase in CD8+ T cells compared to the vehicle controls [57]. Furthermore, the activation of NOTCH1 in CRC triggers CXCR2-dependent neutrophil recruitment to the pre-metastatic niche and promotes an immunosuppressive microenvironment. Targeting CXCR2 in this setting resulted in an increase in CD8+ T cells within the premetastatic niche and a reduction in the metastatic burden. These in vivo data further support potential applications in the minimal residual disease setting of targeting myeloid cells, thereby impeding metastatic seeding. Considering these mechanisms, we believe that MDSCs play significant roles in promoting metastases and worse outcomes in patients with gastrointestinal malignancies.

The GALAXY trial, part of the umbrella CIRCULATE-Japan ctDNA platform protocol, has been very informative in confirming the prognostic and predictive value of ctDNA in patients with stage II to IV CRC [59,60]. These data confirmed the clinical utility of ctDNA analysis for identifying MRD and its impact on patient-specific survival outcomes. GALAXY enrolled 1564 patients, of whom 1039 (the “outcome cohort”) were included in the analysis [60]. The group was divided into three cohorts. The first cohort consisted of patients who tested ctDNA positive following 4 weeks of surgery (*n* = 187), and in whom ctDNA clearance was analyzed during adjuvant chemotherapy. This group was known as the “clearance cohort.” The second cohort consisted of patients who tested ctDNA negative (*n* = 531) 4 weeks after surgery, and the third group was the “dynamics analysis cohort” (*n* = 838), for which outcomes associated with changes in the ctDNA status over the course of the treatment were evaluated. The ctDNA status and the prognostic impact on disease-free survival (DFS) was noted, and patients who tested positive for ctDNA 4 weeks after surgery had an increased risk for recurrence by 11 to 13 times. For patients with stage I–IV disease, the 6-month and 12-month DFS rates for patients with a ctDNA positive status were 62% and 47.5% compared to 96.5% and 92.7% for those who were ctDNA negative, respectively (HR 10.9, *p* < 0.001). For patients with stage II–III disease, the 6-month and 12-month DFS rates were noted to be 73.0% and 55.5% for patients with ctDNA positive statuses compared to 97.8% and 95.2% for those who were ctDNA negative, respectively (HR 13.3, *p* < 0.001) [59]. Lastly, the ctDNA dynamic changes were predictive of disease recurrence; specifically, patients with a baseline ctDNA positivity who remained positive over the course of their adjuvant therapies had a 15.8-fold increased risk of recurrence [60]. In summary, these clinical translational data support the critical need for novel therapeutic approaches for patients who remain ctDNA positive after completing curative intent therapy and further supports the rationale of our current protocol planned Arm B ctDNA cohort (NCT04599140) as well as numerous other MRD protocols that are ongoing and in development [61].

## 7. Conclusions

In conclusion, the targeting of myeloid cells in cancer therapy represents an exciting tumor agnostic immunotherapeutic approach with increasing data supporting MDSCs’ critical roles in multiple tumors [13,39,40,57]. The preclinical data support CXCL3-CXCR2 as a key effector of the *KRAS*-mutated interferon regulatory factor 2 (IRF2) axis and provides an actionable approach to inhibit MDSC infiltration and thereby enable T cell-mediated tumor immunity in CRC [13]. These data highlight a potential role of the CXCL3/CXCR2 axis in impacting the tumor immune microenvironment of advanced MSS CRC, including patients with ctDNA positive CRC post curative intent therapies. Defining a biomarker for CRC patient selection that would predict responses to immunotherapies targeting CXCR2 positive MDSCs represents a notable challenge and warrants ongoing investigation. The CMS4 CRC subtype reflects a unique mesenchymal phenotype with the enrichment of MDSCs and may potentially serve as a predictive biomarker of selection moving forward depending on the success of the therapeutic inhibition of MDSC trafficking. Whether CXCR1/2 inhibition in combination with anti-PD1 clinically modulates the TME by impeding MDSC trafficking and decreasing T regulatory cells, thereby promoting an increased presence of cytotoxic CD8 T cells and tumor death in CRC, remains to be discovered (STOPTRAFFIC-1, NCT04599140). Additionally, subsequent combinatorial approaches will likely be required to adequately modulate the immune microenvironment in MSS CRC in a durable fashion. This approach represents one of the many avenues that are being actively investigated in MSS CRC to allow for the successful deployment of a novel immunotherapy for patients with limited treatment options.

## Figures and Tables

**Figure 1 cancers-15-05484-f001:**
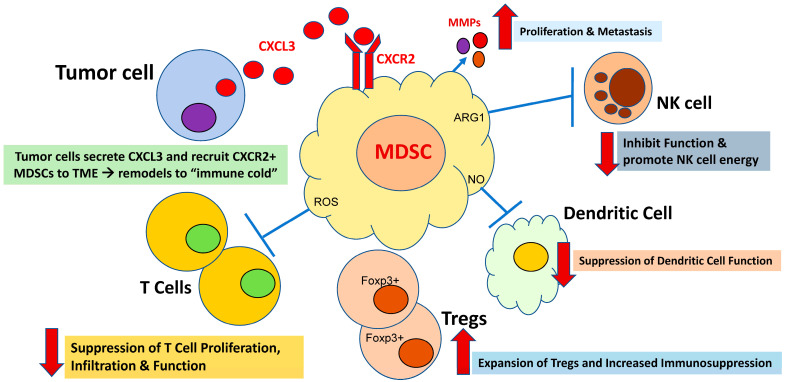
MDSCs’ roles in *RAS*-mutated colorectal cancer; they promote immunosuppressive tumor microenvironment (TME) and progression of metastasis.

**Figure 2 cancers-15-05484-f002:**
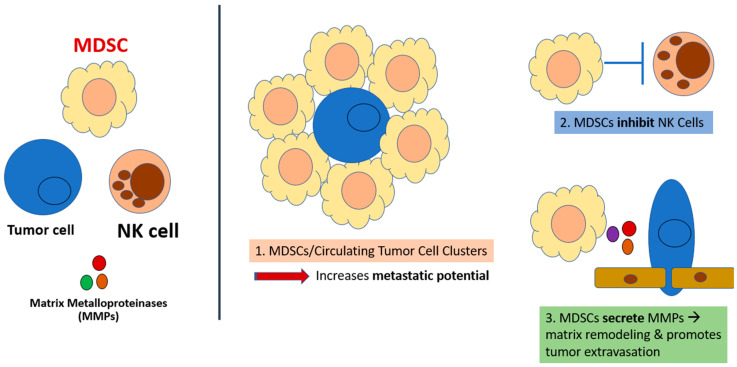
MDSCs roles in the activation of premetastatic niches and minimal residual disease.

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
