# Peer review of "Targeting Myeloid-Derived Suppressor Cell Trafficking as a Novel Immunotherapeutic Approach in Microsatellite Stable Colorectal Cancer"

_cancers, 2023, doi:10.3390/cancers15225484_

Round 1

Reviewer 1 Report

Comments and Suggestions for Authors

The author presents a relevant review of the literature in the context of KRAS mutant CRC and the role of CXCR2 chemokines in generating MDSC infiltration. The work is based extensively on in vivo studies particularly from MD Anderson (De Pinho). There are some references missed in this context - Jackstadt et al Cancer cell 2019. iKAP model is referred to repeatedly and this mechanism could be broadened and could include previous studies that have been performed in other KRAS mutant cancers such as lung and pancreas as the mechanism is likely to be tumour agnostic. Mention should be made of recent work showing CD40 agonists activating different neutrophil phenotypes and it would seem sensible to consider neutrophils and macrophages in terms of recent single cell sequencing data showing myriad subtypes. 

This preclinical work in this area has not translated successfully - PRIMUS 003 for instance in pancreas. A section on how to determine patients for therapy would be useful. 

Author Response

Reviewer 1 Comments and Author Responses

The author presents a relevant review of the literature in the context of KRAS mutant CRC and the role of CXCR2 chemokines in generating MDSC infiltration. The work is based extensively on in vivo studies particularly from MD Anderson (De Pinho).

There are some references missed in this context - Jackstadt et al Cancer cell 2019.

            Added Jackstadt reference and brief summary to the premetastatic niche section of targeting CXCX2+ neutrophils / CMS4. -page 7

iKAP model is referred to repeatedly and this mechanism could be broadened and could include previous studies that have been performed in other KRAS mutant cancers such as lung and pancreas as the mechanism is likely to be tumour agnostic.

            Incorporated additional preclinical model references (iKRAS in pancreas and KRAS G12D lung model) to the KRAS models and TME section.  – page 5

Mention should be made of recent work showing CD40 agonists activating different neutrophil phenotypes and it would seem sensible to consider neutrophils and macrophages in terms of recent single cell sequencing data showing myriad subtypes. 

            Added CD40 agonist work referencing this point – page 6

This preclinical work in this area has not translated successfully - PRIMUS 003 for instance in pancreas. A section on how to determine patients for therapy would be useful. 

            Added sentences regarding status of clinical biomarker/patient selection barriers in the conclusions.  Considering CMS4 CRC is suggestive of a myeloid rich environment this may serve as a notable biomarker for future selection if the regimen is active.  – page 8

Reviewer 2 Report

Comments and Suggestions for Authors

This is an excellent and extensive review dealing with current therapies against CCR, notably those that not associated to microsatellite instability.

The author concludes “Whether CXCR1/2 inhibition in combination with anti- PD1 modulates the TME in CRC by impeding MDSC trafficking and decreasing T regulatory cells to the TME ultimately results in an increased presence of cytotoxic CD8 T cells promoting tumor kill in CRC remains to be discovered.

Perhaps, it would be to precise how could be clinical trials?

Detail point:

A large part of the introduction deals with several (sub)-types of haematopoietic cells and their role(s) in cancer +/- defense. Perhaps, and since the review is very interesting, for non-specialists in immunology, I would appreciate to see a cartoon reminding the reader the main oncogenic events known up to date (even as Sup. Mat.). This could perhaps help to shorten a bit the introduction section ?

Author Response

Reviewer 2 Comments and Author response

This is an excellent and extensive review dealing with current therapies against CCR, notably those that not associated to microsatellite instability.

The author concludes “Whether CXCR1/2 inhibition in combination with anti- PD1 modulates the TME in CRC by impeding MDSC trafficking and decreasing T regulatory cells to the TME ultimately results in an increased presence of cytotoxic CD8 T cells promoting tumor kill in CRC remains to be discovered.

Perhaps, it would be to precise how could be clinical trials?

The clinical trial is currently ongoing at MD Anderson Cancer Center evaluating a CXCR1/2 inhibitor and anti-PD-1 in refractory colon cancer as well as in ctDNA positive settings and has just started dose expansion.  Added the clinicaltrials.gov NCT reference to the conclusion and in the body of the text.  – page 9

Detail point:

A large part of the introduction deals with several (sub)-types of haematopoietic cells and their role(s) in cancer +/- defense. Perhaps, and since the review is very interesting, for non-specialists in immunology, I would appreciate to see a cartoon reminding the reader the main oncogenic events known up to date (even as Sup. Mat.). This could perhaps help to shorten a bit the introduction section?

To try and cover all the various immune cell subtypes would be outside of the scope of this review as we are focusing primarily on the MDSCs role.  I did add Figure 1 to highlight the various roles MDSCs are playing in this setting. 

Round 2

Reviewer 1 Report

Comments and Suggestions for Authors

Comprehensive review and improved since first version - publish